# Two-Dimensional Magnetotelluric Parallel-Constrained-Inversion Using Artificial-Fish-Swarm Algorithm

**Zuzhi Hu [1]** , **Yanling Shi [1]**, **Xuejun Liu [1]**, **Zhanxiang He [2,3,\*]** , **Ligui Xu [1]**, **Xiaoli Mi [1] and Juan Liu [1]**

1    BGP Inc., CNPC, Zhuozhou 072751, China
2    Guangdong Provincial Key Laboratory of Geophysical High-Resolution Imaging Technology, SUSTech, Shenzhen 518055, China
3    Department of Earth and Space Science, SUSTech, Shenzhen 518055, China
\*    Correspondence: hezx@sustech.edu.cn

**Abstract:** An important way to improve the resolution of electromagnetic exploration is by using known seismic and logging data. Based on previous work, 2D magnetotelluric (MT) parallel-constrained-inversion, based on an artificial-fish-swarm algorithm is further developed. The finite-difference (FD) method with paralleling frequency is used for 2D MT-forward-modeling, to improve computational efficiency. The results of the FD and finite-element (FE) methods show that the accuracy of FD is comparable to FE in the case of suitable mesh-generation; however, the calculation speed is ten times faster than that of the FE. The artificial-fish-swarm algorithm is introduced and applied to parallel-constrained-inversion of 2D MT data. The results of the synthetic-model test show that the artificial-fish-swarm-inversion based on paralleling forward can recover the model well and effectively improve the inversion speed. The processing and interpretation results of the field data are verified by drilling, which shows that the proposed inversion-method has good practicability.

**Keywords:** finite-difference method; artificial fish swarm; magnetotelluric; parallel-constrained-inversion

## 1. Introduction

In many areas of existing exploration, seismic data has high resolution in the middle-shallow layer, but the deep structure in a complex region is unclear, and electromagnetic data can provide an important supplement. Furthermore, it is an important way to improve the resolution of electromagnetic exploration using the known seismic, geological, and well-logging data [1–8]. Several workshops on joint inversion have been held in recent SEG annual meetings, and constrained or joint inversion has become an important technique to improve the resolution of geophysical data in many other technical sessions.

Using joint inversion, a model combining data obtained using different methods can be established, and the mutual constraints and complementary advantages of these methods can realize a fine mapping of structures [9–13]. Brown et al. [14] proposed that constraining controlled-source-electromagnetic (CSEM) inversion with full-seismic-waveform-inversion results can improve the resolution of the results. Vieira da Silva et al. [15] applied full-waveform-inversion of acoustic velocity to three-dimensional constrained inversion of marine electromagnetic-data. They used comprehensive velocity-field information to obtain a more reliable initial model for the three-dimensional inversion of CSEM data, to reduce the uncertainty of CSEM-inversion results and the influence of transceiver distance. Hu et al. [16] developed a parallel magnetotelluric (MT)-constrained-inversion algorithm based on fast-simulated-annealing, which can effectively reveal deep high-resistivity anomalies based on seismic-horizon interpretation constraints. Zhou et al. [17] realized the joint inversion of MT and seismic data with cross-gradient constraints, and improved the accuracy of MT-inversion results. Hu et al. [18] developed a one-dimensional MT-artificial-fish-swarm constrained-inversion, based on seismic-horizon constraints, which

can provide geological interpreters with more information on deep lithological changes. Li et al. [19] proposed a three-dimensional resistivity-inversion-method, based on prior constraints of the spatial shape of anomalous bodies, which suppresses the multi-solution problem of three-dimensional resistivity-inversion. Bergman et al. [20] used the structurally-constrained-inversion method to study the joint processing of seismic and electromagnetic data. The interpretation of the horizon structure from seismic data provided constraints for electromagnetic-data inversion. The method was applied to time-lapse monitoring of $CO_2$-gas-injection stations. Constrained-resistivity-inversion produced clearer images of the interface between the caprock and reservoir. Shi et al. [21] applied MT-constrained-inversion by shallow logging and seismic data to reveal the existence of a rift-type faulted basin in the deep part of the central Sichuan Basin; Chen et al. [22] took MT and seismic joint-inversion as an example, proposed a new framework of joint inversion, designed a typical basalt model to verify the method, and solved the problem of unclear underlying-strata morphology through the joint inversion of resistivity and velocity. With the development of constrained- and joint inversion processing technology, the ability of electromagnetic exploration methods to solve geological problems will be improved significantly. Dong et al. [6] realized the pseudo-2D-joint-inversion calculation of TEM and CSAMT with well-logging constraints. Compared with either TEM or CSAMT, joint inversion with well-logging constraints has a significantly better capability of reflecting water abundance in rock formation and faults.

Based on the work of Hu et al. [18], this paper further develops the parallel-constrained-inversion of 2D MT data to improve the accuracy of electromagnetic inversion. Compared with the traditional gradient-inversion method [23–26], the artificial-fish-swarm method only needs to calculate the output value of the objective function, without calculating gradient and sensitivity information. It has the characteristics of simplicity, robustness, parallelism, and fast convergence. Firstly, the finite-difference method of 2D MT-forward-modeling is introduced, and we take parallel-forward-modeling to improve computational efficiency. The forward result is compared with that of the finite-element method, to verify the correctness of the method. Then, the artificial-fish-swarm inversion-method is introduced, which is applied to the parallel-constrained-inversion of 2D MT data. The results of synthetic data are compared with the quasi-2D inversion results. Finally, the field MT data in the Sichuan Basin are processed, and the results are discussed.

## 2. Parallel-Forward-Modeling of 2D MT Data

At present, 2D forward-modeling of MT data is mainly based on the finite-element method (FE) and finite-difference (FD) method [27–41]. Because the triangular unit subdivision is highly adaptable to terrain and complex anomalous-body simulation, and the accuracy of the quadratic function interpolation is higher than that of the linear interpolation in a unit, the use of the FE method of triangular unit subdivision for the MT-data-forward modeling [29], is the most widespread. Smith and Booker [31] applied the FD method to calculate the forward response of MT. Theoretical model studies show that the FD method can obtain the highest precision response of complex models with suitable mesh, and its computation time is distinctly shorter than that of the conventional FE method. Therefore, we choose the FD method for 2D MT-forward-modeling [31].

$$\mathbf{A} \cdot \mathbf{x} = \mathbf{b} \qquad (1)$$

where $\mathbf{A}$ is the symmetric sparse complex-coefficient matrix; $\mathbf{x}$ is a column vector of grid points or component of $E_x$ or $H_x$; and the right-hand vector, $\mathbf{b}$, is a vector related to frequency and resistivity of grid elements. After solving Equation (1), $H_y$ and $E_y$ can be solved by Maxwell's equations. Apparent resistivity and phase at surface observation stations can be obtained from the calculated field values. The apparent resistivity and phase of *TE* and *TM* modes are, respectively, as in [31]

$$\begin{cases} \rho_{TE} = \frac{1}{\omega\mu_0}\left|\frac{E_x}{H_y}\right|^2, \theta_{TE} = Arg\left(\frac{E_x}{H_y}\right) \\ \\ \rho_{TM} = \frac{1}{\omega\mu_0}\left|\frac{E_y}{H_x}\right|^2, \theta_{TM} = Arg\left(\frac{E_y}{H_x}\right) \end{cases} \tag{2}$$

where $\omega$ is the angular frequency and, $\mu_0$ is the magnetic permeability in vacuum.

### 2.1. Basic Approach of Parallel-Forward-Modeling

Although the computational speed of the FD method is faster than that of the FE method, its computational efficiency can still be improved. Parallel design is one of the methods to improve computational efficiency. Because 2D-forward-modeling of MT data calculates each frequency separately and the corresponding electromagnetic-field value of each frequency is independent of the others, adopting a parallel algorithm for forward modeling is suitable.

We use the principal-subordinate pattern for programming, which has a main process and many subordinate processes. The main process takes charge of assigning and sending tasks to each process, reclaiming and outputting the computed results. The subordinate process takes charge of the task, and sends the computed results back to the main process [42,43].

The main steps of two-dimensional parallel-forward-modeling are as follows: (1) initialization of MPI parameters; (2) main process reads parameters such as 2D-resistivity model files, grid data, and frequency data; (3) main process broadcasts all parameters to each subordinate process, and assigns calculation tasks of each subordinate process, according to frequency; (4) each subordinate process receives calculation tasks and performs forward calculation, and returns the calculation results to the main process; (5) the main process receives the calculation results of each subordinate process, and outputs the apparent resistivity and phase data.

### 2.2. Parallel-Forward-Verification and Comparison

To verify the correctness of the 2D FD-parallel-forward program, the results are compared with those obtained by the FE method [29,30]. According to the geological model, a complex 2D geo-electric model is made, as shown in Figure 1. The length of the model section is 19.6 km, the depth is 13.6 km, the distance between the measuring stations is 0.1 km, and the mesh of forward calculation is divided into 220 × 134. The frequency is divided into 40 frequency points which are uniformly distributed in the logarithmic domain between 0.001 and 320 Hz. The two forward methods adopt the same parameters. On 32 threads with 16 cores and a 3.1 GHz computer workstation, 1, 5, 10, and 20 threads are used to parallel forward. Figure 2 is the comparison of apparent-resistivity sections of the TE and TM modes from the FD method and FE method, respectively. Figure 3 shows the relative error percentage of apparent resistivity in the TE and TM modes of the 2D FD and FE methods of forward modeling. It can be seen that the maximum relative error of apparent resistivity of the two forward methods is between −3% and 2.5%, mainly in the low-frequency band, and the relative error of most frequency bands is within ±0.5%. This shows that when the model is reasonably divided, the forward precision of the FD method is the same as that of the FE method, which can meet the forward and inversion requirements of 2D MT.

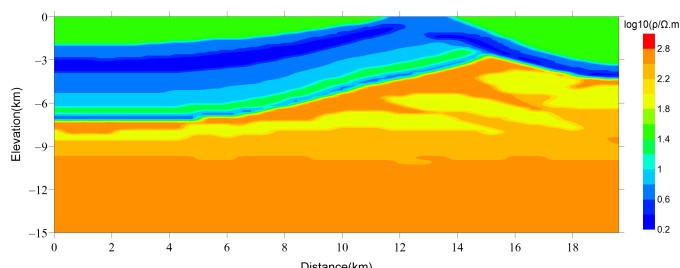

**Figure 1.** 2D resistivity model.

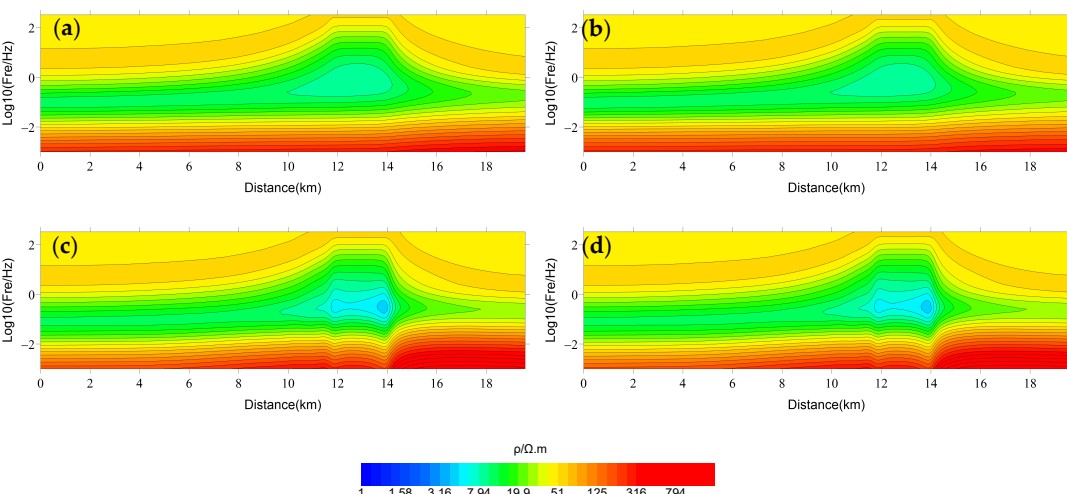

**Figure 2.** 2D comparison of apparent-resistivity sections of TE (upper) and TM (lower) mode by different forward-methods. (**a**,**c**) are calculated using the FD method; (**b**,**d**) are calculated using the FE method.

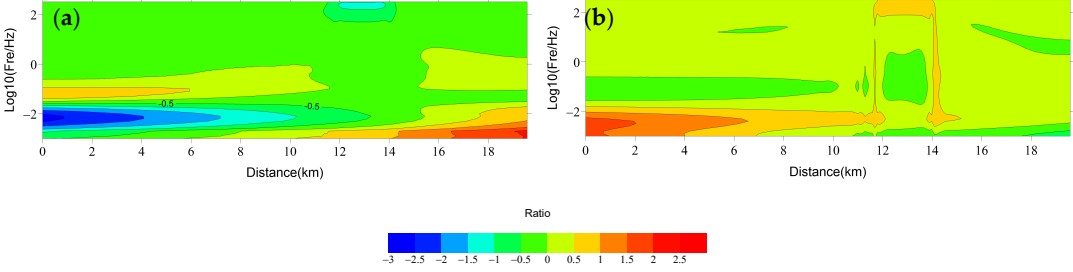

**Figure 3.** Relative error percentage of apparent resistivity between 2D FD and FE forward method. TE mode (**a**); TM mode (**b**).

For the model in Figure 1, the forward mesh is divided into $220 \times 134$ with 40 frequencies, and the FD and FE forward-modeling of different CPUs is performed within the same parameters. Figure 4a is a comparison of 2D forward-parallel-computing-time. From Figure 4, it can be seen that when the FE method is used for serial calculation, the time used is 666.9 s, while 44.9 s is used for the FD method. Under the same conditions, the calculation speed of the FD method is nearly 15 times faster than that of the FE method. As shown in Figure 4b, when 20 CPUs are used for parallel computation, the computational time of the FE method and FD method is reduced to 52.1 s and 4.5 s, respectively, and the parallel scaling is 12.8 and 9.98, respectively. By using the FD method and parallel processing, the forward calculation time is reduced from several hundred seconds to several seconds, and the speed is increased by tens of times, which provides a good basis for the non-linear MT-inversion.

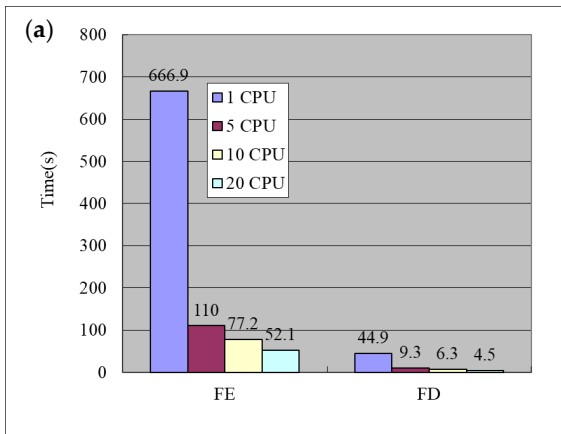
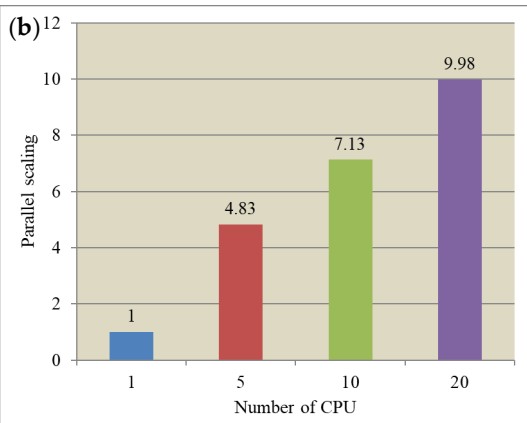

**Figure 4.** Comparisons of 2D forward-parallel-computing time (**a**) and parallel scaling (**b**) of different CPUs between FD and FE methods.

## 3. Constrained Inversion of Artificial-fish-swarm algorithm

The specific details about the basic implementation of the artificial-fish-swarm algorithm are referred to in the literature [18,44]. It includes the following five aspects: model selection, preying, swarming, following, and the objective function.

### 3.1. Model Selection

For a given artificial-fish-swarm, namely the constrained-model space of MT data $\left[M^{\min}, M^{\max}\right]$, which consists of resistivity and thickness. The initial model is generated by the uniform probability distribution in the model space, to simulate the random behavior of the artificial fish. The initial artificial-fish-swarm parameters are as follows:

$$m_i^0 = m_i^{\min} + u_i * (m_i^{\max} - m_i^{\min}) \tag{3}$$

where $m_i^0$ is the $i^{th}$ parameter of the initial model, $M$, $m_i^{\min}$ and $m_i^{\max}$ are the low limit and upper limit, $m_i$ respectively. $u_i$ is a random number, generated from the uniform distribution between 0 and 1.

### 3.2. Preying Behavior

Preying behavior is a basic biological behavior of fish that tends to move toward the food. Generally, the fish perceives the concentration of food in water to determine the movement by vision or sense, and then chooses the tendency. Assuming the current state for artificial fish, $i^{th}$, is $M_i$, which is the model parameter of inversion, we select a state $M_j$, randomly in its visual distance,

$$M_j = M_i + u_v * V \tag{4}$$

where $u_v$ is a random number generated between 0 and 1. $V$ is the visual length of the artificial fish. If the objective function value $M_j$ is greater than $M_i$, it goes forward a step in this direction, in accordance with the Formula (5). Otherwise, select a state $M_j$ randomly again, and judge whether it satisfies the forward condition.

$$M_i^{t+1} = M_i^t + u_p * S * \frac{M_j - M_i^t}{\left\|M_j - M_i^t\right\|} \tag{5}$$

where $S$ is the searching step of the artificial fish.

### 3.3. Swarming Behavior

The fish will assemble in groups naturally in the moving process, which is a kind of survival habit to guarantee the existence of the colony and avoid danger. For the artificial-fish-swarm algorithm, each fish follows two principles. One is that each fish should try to move to the neighboring fish center. The other is that the fish should not be very crowded.

Assume that $M_c$ is the center position and $N_v$ is the number of its companions in the current neighborhood. If $E_c / N_v > \delta * E_i$, which means the companion center has more food and is not very crowded, the artificial fish goes forward a step towards the companion center.

$$M_i^{t+1} = M_i^t + u_s * S * \frac{M_c - M_i^t}{\left\| M_c - M_i^t \right\|} \tag{6}$$

### 3.4. Following Behavior

In the moving process of the fish swarm, when a single fish or several fish find food, the neighborhood partners will follow, and reach the food quickly. This is the following behavior of fish.

We calculate the number of fish in the neighborhood and the best objective function on the basis of what is seen. If the companion state has a higher food-concentration and the surroundings are not very crowded, the fish goes forward a step to the companion.

$$M_i^{t+1} = M_i^t + u_f * S * \frac{M_k - M_i^t}{\left\| M_k - M_i^t \right\|} \tag{7}$$

### 3.5. Objective Function

The objective function for 2D MT-inversion is chosen as [16]

$$\begin{aligned}
\Delta E_m = & \sum_{j=1}^{MS} \sum_{i=1}^{NS} \left\{ \left[ 1 - \rho_{ij}^{TEcal} / \rho_{ij}^{TEobs} \right]^2 + \left[ 1 - \varphi_{ij}^{TEcal} / \varphi_{ij}^{TEobs} \right]^2 \right\} \\
& + \sum_{j=1}^{MS} \sum_{i=1}^{NS} \left\{ \left[ 1 - \rho_{ij}^{TMcal} / \rho_{ij}^{TMobs} \right]^2 + \left[ 1 - \varphi_{ij}^{TMcal} / \varphi_{ij}^{TMobs} \right]^2 \right\}
\end{aligned} \tag{8}$$

where $MS$ is the total MT sounding stations, and $NS$ is the number of frequencies. $\rho_{ij}^{TEcal}$, $\rho_{ij}^{TMcal}$, $\varphi_{ij}^{TEcal}$ and $\varphi_{ij}^{TMcal}$ are the calculated apparent resistivity and phase of the TE mode and TM mode, respectively. $\rho_{ij}^{TEobs}$, $\rho_{ij}^{TMobs}$, $\varphi_{ij}^{TEobs}$ and $\varphi_{ij}^{TMobs}$ are the observed apparent resistivity and phase of the TE mode and TM mode, respectively.

In the process of calculating the objective function, there is a part which depends on MT-forward-calculation. For the 2D artificial-fish-swarm algorithm, a large number of 2D MT-forward-calculations are needed. Therefore the improvement of the computational speed of 2D forward-modeling is the key to the practicability of this method.

### 3.6. The Basic Idea of Parallel Inversion

There is a relatively large error in processing 2D data with a 1D-inversion program, especially since the underground media shows obvious 2D geological characteristics. Therefore, we develop an inversion of 2D MT data, based on an artificial-fish-swarm algorithm, to overcome the false anomaly caused by 1D inversion. To improve the inversion speed, we adopt the 2D FD method for forward modeling, and take advantage of the MPI parallel-design strategy [42,43] to realize parallel-artificial-fish-swarm constrained-inversion of 2D MT data.

The main steps of 2D MT parallel-constrained-inversion of the artificial-fish-swarm are as follows: (1) initialization of MPI parameters; (2) the main process reads 2D MT observed data, 2D constrained-model files, and other parameters; (3) the main process carries out artificial-fish-swarm inversion; (4) in the process of the artificial-fish-swarm-inversion, many 2D MT FD-forward-calculations are involved, and the main process broadcasts all the parameters to each subordinate process; the forward-calculation tasks of each subordinate process are allocated according to the number of frequency; (5) each subordinate process receives forward-calculation tasks, and returns the results to the main process after calculation; (6) the main process calculates the misfit, outputs the inversion results, the forward apparent-resistivity and phase data. Figure 5 shows the parallel computational flow of 2D MT parallel-constrained-inversion for the artificial fish-swarm. Next, we use synthetic models to compare the results of quasi-2D and 2D inversion.

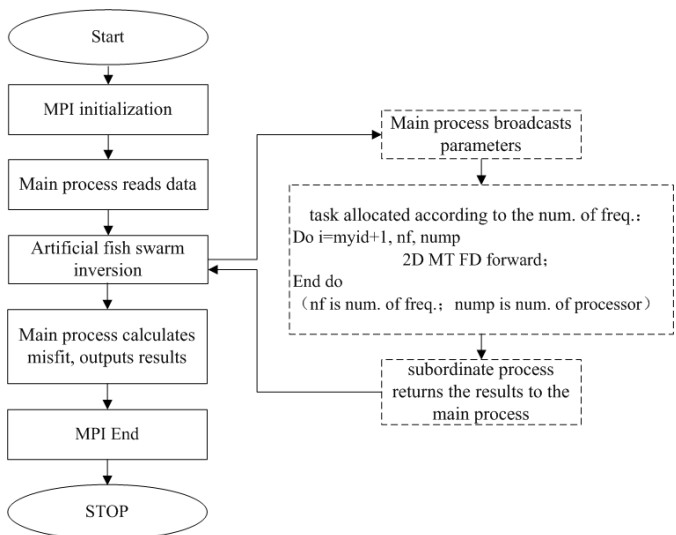

**Figure 5.** Brief diagram of 2D MT parallel-constrained-inversion for the artificial fish-swarm.

## 4. Synthetic-Model Tests

### 4.1. Quasi-2D Parallel Inversion

Figure 6a is a 2D resistivity model with a section length of 10 km, a depth of 6 km, and a green-background resistivity of 100 $\Omega \cdot$ m. An anomalous body with a low resistivity of 10 $\Omega \cdot$ m is embedded at a depth of $-1$ to $-2$ km at 4–6 km in the horizontal direction. The site interval of 2D MT-modeling is 0.2 km, and the number of sites is 51.

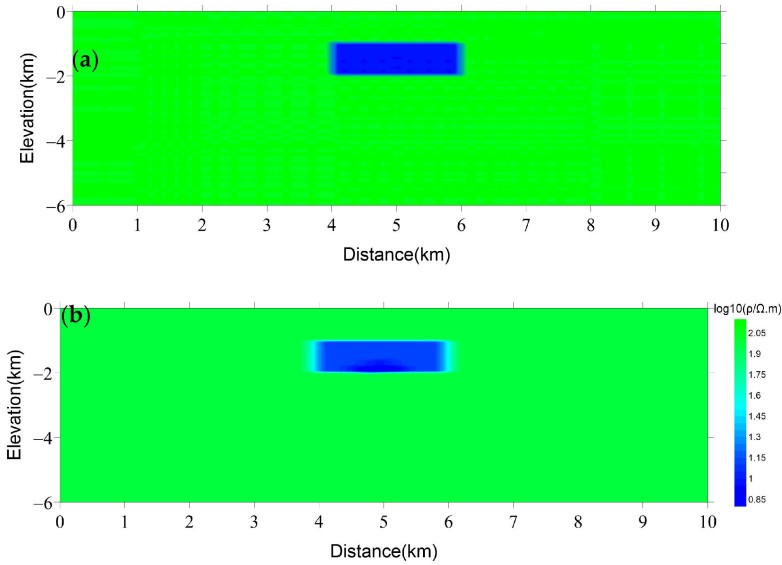

**Figure 6.** 2D-resistivity model (**a**) and quasi-2D-inversion-resistivity section (**b**).

We perform the artificial fish-swarm constrained-inversion of the TM data by constraining the shape of the anomalous body, to obtain the resistivity. The inversion data has 40 frequencies between 0.001 and 320 Hz. The constrained-model parameters for inversion are ±30% of the real model; the maximum iteration number is 50. After 50 iterations, the resistivity section of the quasi-2D artificial fish-swarm constrained-inversion is shown in Figure 6b, and the resistivity value of the low-resistivity body is well recovered. Figure 7 is the misfit curve of a single station. Except for the misfit on the anomalous body, which is slightly larger, the misfit of the other stations is less than 0.01, which fits well.

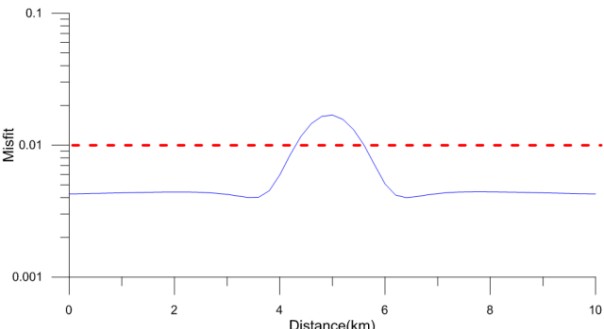

**Figure 7.** The misfit curve of a single station. The red dotted line is the threshold value of 0.01.

### 4.2. 2D Parallel Inversion

The test model also uses the 2D-resistivity model shown in Figure 6a. We perform artificial fish-swarm constrained-inversion of the TM apparent-resistivity and phase data, respectively, by constraining the shape of the anomalous body to obtain the resistivity. The grid of the model is discretized to $65 \times 31$; the parameters are the same as above.

The processes 1, 2, 4, 8, 10, and 20 are used for parallel inversion, separately. The computations were stopped after 50 iterations. The results of parallel inversion with 10 CPUs are shown below. Figure 8 shows the misfit curve of 2D TM-mode inversion with 50 iterations, in which the misfit is less than $1.0 \times 10^{-4}$. Figure 9a shows the inversion results of the TM apparent-resistivity data. Figure 9b is the initial resistivity-value distribution of the anomalous body. Figure 9c shows the final resistivity-value distribution for the inverted anomalous body. We find that by using the maximum probabilistic estimate, the resistivity value of the inverted anomalous body is finally distributed around 10, which is very close to the true value. This indicates that the 2D inversion program is valid and feasible. Figure 10 shows the apparent-resistivity section of the model and the calculated apparent-resistivity section of the 2D-inversion result, which fit each other very well.

Figure 11a is the time comparison of 2D MT-parallel-constrained-inversion with different numbers of CPU. The inversion time is the longest with 1 CPU, requiring 5861 s. With the increase in the number of CPUs, the inversion time is gradually reduced. When the number of CPUs reaches 20, the inversion time is only 384 s, which, when sped up, reaches 15.26 times. It can be seen that 2D MT-inversion efficiency is obviously improved with the parallel-inversion algorithm.

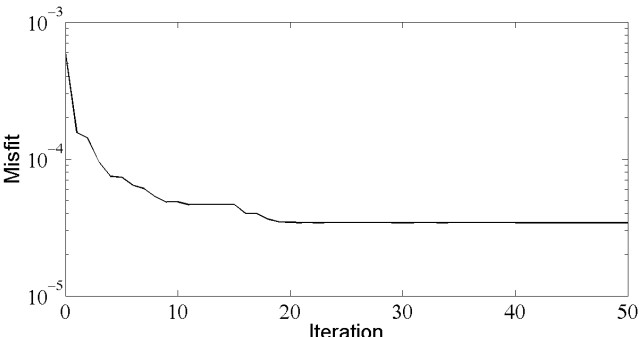

**Figure 8.** The misfit curve of 2D TM-mode inversion.

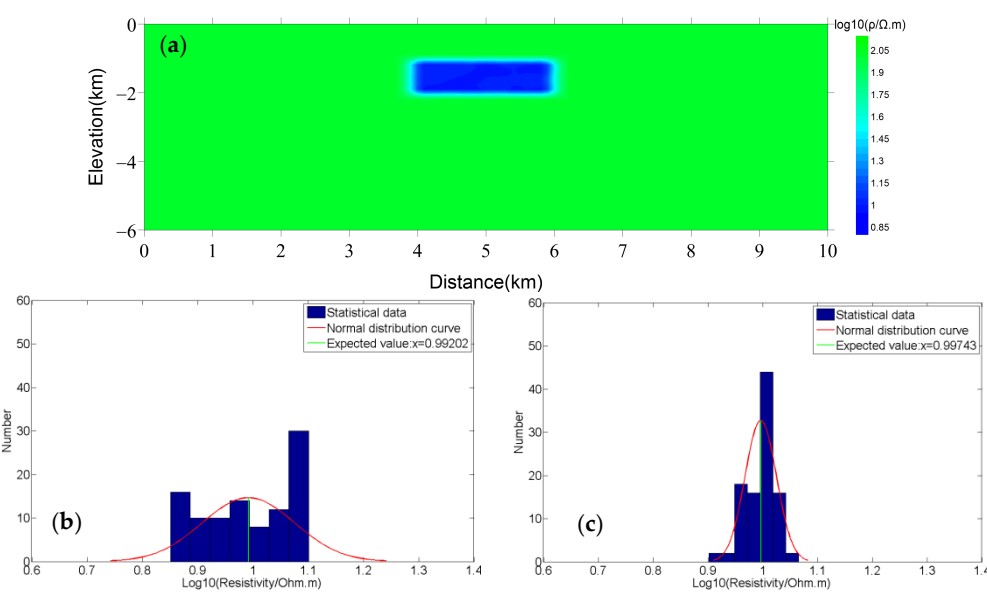

**Figure 9.** Individual inversion results of apparent-resistivity data in TM mode (**a**); abnormal resistivity-distribution of initial inversion (**b**); abnormal resistivity-distribution of final inversion (**c**).

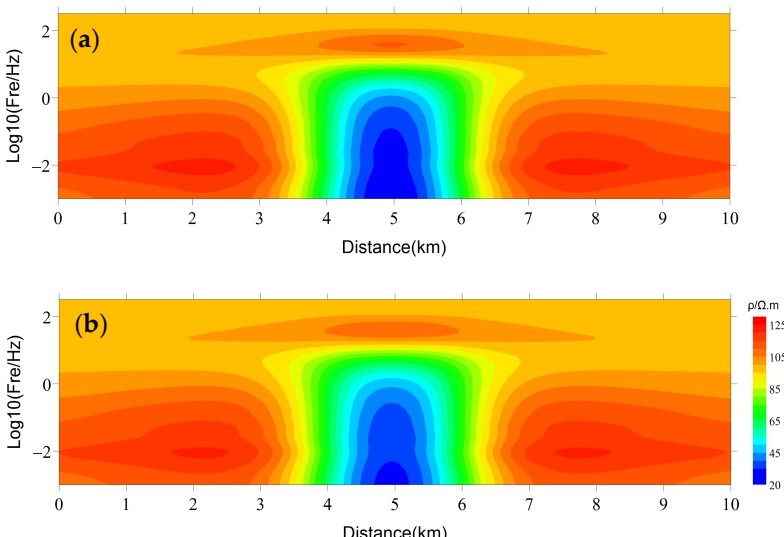

**Figure 10.** Apparent-resistivity sections of the true model (**a**) and the model (**b**) from 2D inversion.

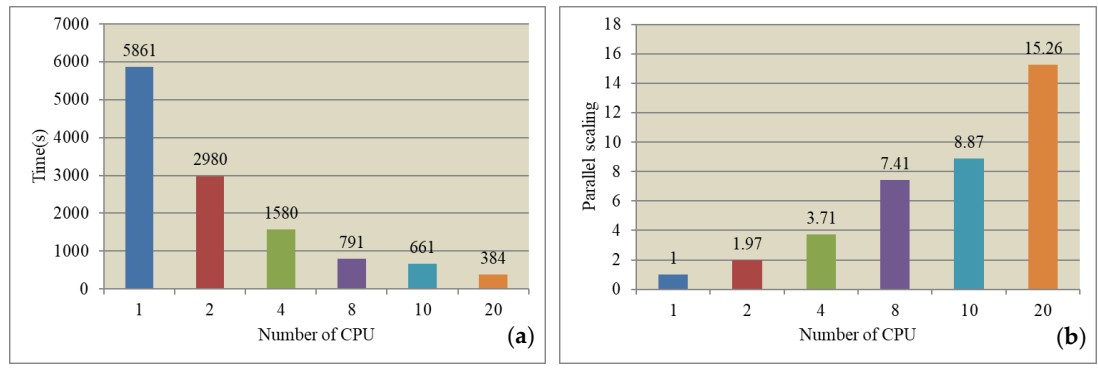

**Figure 11.** Time (**a**) and parallel-scaling (**b**) comparisons of 2D MT-parallel-constrained-inversion.

## 5. Field-Data Results

The field MT-data are acquired in the GL area of the Central Sichuan Basin. Previous drilling and seismic exploration show that the distribution of Sinian oil and gas in the

Central Sichuan Basin is closely related to the development of a deep rift in the Sichuan Basin [21]. To further understand the situation of the Sinian rift development in the GL area, the MT survey lines, L1, are deployed at the same station, as shown in Figure 12.

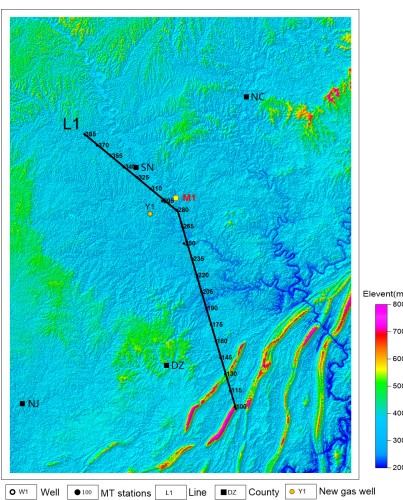

**Figure 12.** Location of survey line in the study area.

### 5.1. Characteristics of Formation Resistivity

The statistical resistivity of the formation rocks is analyzed using the seven wells with whole resistivity data in the study area and the adjacent area. The resistivity above the Upper Triassic strata is low in the study area; the Jialingjiang formation of the Lower Triassic to the Leikoupo formation of the Middle Triassic is a secondary high-resistivity layer; the Feixianguan formation of the Lower Triassic to the Middle Permian shows low resistivity; the Lower Permian strata has high resistivity; the Cambrian and Ordovician strata are characterized by medium to low resistivity; the Upper Sinian strata has high resistivity.

According to the previous research results, if there is an early rift in the study area, it must have developed in the Lower Sinian strata. Combined with the field-measured data in the study area and the adjacent area, the resistivity of the Lower Sinian and basement are calculated. The deep fault-depression has various sedimentary lithologies, including sandstone, shale, mudstone, and volcanic rocks. In addition, there is a set of moraine deposits in this area. According to the statistical results, the sediments in the early fault depression are characterized by low resistivity.

### 5.2. Constrained-Inversion Results

Conventional and constrained 2D MT-inversion for Line L1 is implemented, as shown in Figure 13. The length of Line L1 is 161 km, with 299 stations, and the station spacing is approximately 0.5 km. The grid for 2D MT-inversion is $318 \times 121$, and data from 19 frequencies, from 0.001 to 320 Hz, are inverted.

Before inversion, the MT data have been denoised [45,46]. Logging data and seismic-interpretation profiles are used to build the constrained model. The resistivity range of each layer during inversion is specified according to the physical-property statistics. The constrained-model parameters for inversion are $70-130\%$ of the real model; the maximum iteration number is 30; the number of artificial fish is 10; the number of attempts is 2; the number of initial searching steps is 10; the visual length is 10; the congestion factor is 0.1, and the desired misfit of the objective function is set to $\Delta E < 10^{-4}$. Ten processes are used for parallel inversion. The computation is stopped normally stopped after 30 iterations.

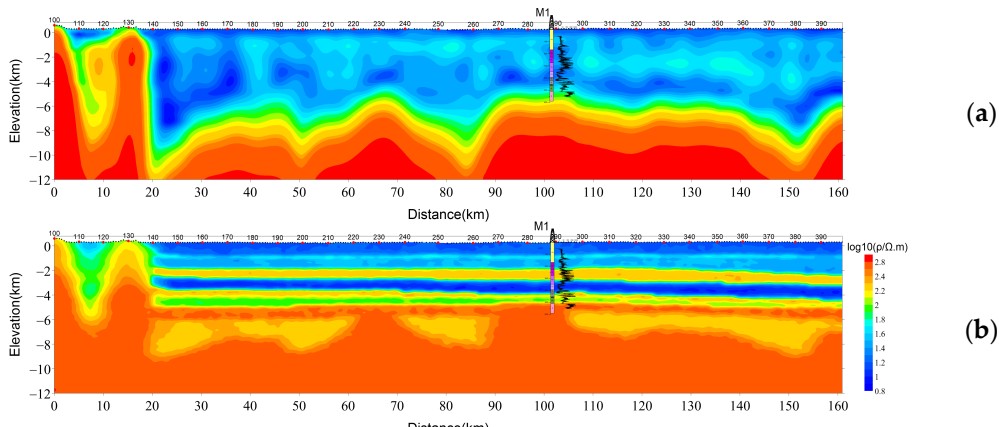

**Figure 13.** Conventional-2D-inversion result (**a**) and constrained-inversion result (**b**) of Line L1.

Figure 14 shows the apparent-resistivity sections of measured data and calculated data from the constrained-inversion results for the TM mode. It can be seen from Figure 14 that the response of 2D-constrained inversion fits well with the measured data in most frequency bands. The fit in some areas can be improved; for example, in the middle- and low-frequency bands of the profile, between 10 and 20 km, the fit is poor, and it is necessary to modify the constraint of the geological model and the range of the constraint parameters.

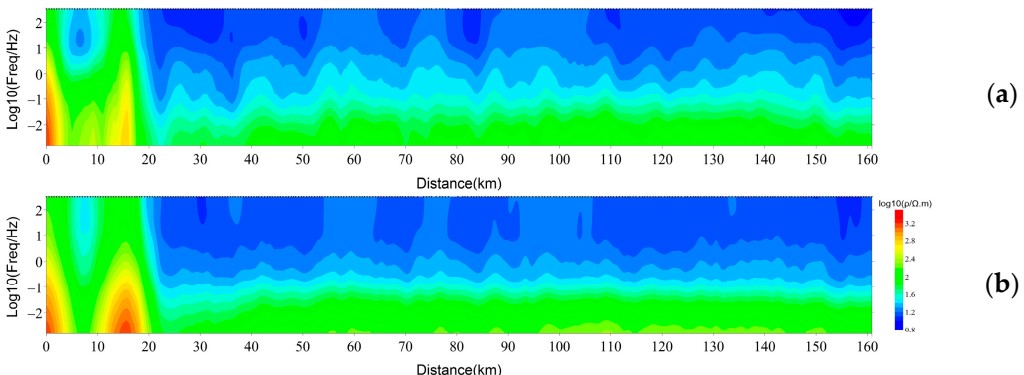

**Figure 14.** Apparent-resistivity sections of measured data (**a**) and calculated data from constrained-inversion result for TM mode (**b**).

In 2016, the new well, Y1, was drilled, after this research. Well Y1 is located in the rift area near the fault depression area of the Nanhua rift, which is seen as a favorable area for oil and gas accumulation, from the MT and gravity data. The interpretation was confirmed by the test results of well Y1; 1.15 million m$^3$ of gas is produced daily.

## 6. Conclusions

We propose a parallel-constrained-inversion of 2D MT data based on the artificial-fish-swarm algorithm. The method is applied to the inversion of the field data in the Sichuan Basin, and the favorable targets of the deep rift and the Nanhua period in the Middle Sichuan Basin have been found and tested by drilling. The study results show that:

(1)  the accuracy of FD is comparable to FE; however, the calculation speed is ten times faster than that of FE;

(2)  the artificial-fish-swarm-inversion based on paralleling forward can effectively improve the inversion speed. For parallel 2D MT-inversion, the maximum acceleration is 15, by using 20 CPUs;

(3)  the proposed inversion-method has good practicability.

**Author Contributions:** Conceptualization, Z.H. (Zhanxiang He) and X.L.; methodology, software, and visualization, Z.H. (Zuzhi Hu); formal analysis, investigation and resources, Y.S.; data curation, X.M.; project administration, L.X.; writing—original draft preparation, Z.H. (Zuzhi Hu), Y.S. and J.L.; writing—review and editing, Z.H. (Zhanxiang He) and L.X. All authors have read and agreed to the published version of the manuscript.

**Funding:** This research was funded by the Science Research and Technology Development Projects of CNPC (2021DJ3706) and National Natural Science Foundation of China (No. 41874085) and Guangdong Provincial Key Laboratory of Geophysical High-resolution Imaging Technology (2022B1212010002) and Shenzhen Science and Technology Project (KQTD 20170810111725321).

**Institutional Review Board Statement:** Not applicable.

**Informed Consent Statement:** Not applicable.

**Data Availability Statement:** Not applicable.

**Conflicts of Interest:** The authors declare no conflict of interest.

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
