# Peer review of "Two-Dimensional Magnetotelluric Parallel-Constrained-Inversion Using Artificial-Fish-Swarm Algorithm"

_magnetochemistry, doi:10.3390/magnetochemistry9020034_

Round 1

Reviewer 1 Report

Dear authors

Introduction

In order to improve the resolution of the electromagnetic exploration, the authors develop a 2D magnetotelluric (MT) parallel, constrained inversion based on artificial fish swarm algorithm. To improve computational efficiency, they carry out a comparative study between Finite Differences (FD) and Finite Elements (FE).The results are comparable, despite this the FD calculation speed is ten times faster than that of the FE. Finally, the method is applied in a certain region of China.

  • Key results: The highlight of the work is to propose a parallel constrained inversion of 2D MT data based on artificial fish swarm algorithm.

  • Validity: The manuscript has no flaws that would prohibit its publication.

  • Originality and significance: The conclusions are original in the sense that they propose a new algorithm based on paralleling forward which can effectively improve the investment speed.

  • Data & methodology: Data not applicable. Regarding the methodology, the authors clearly refer to previous works that they or other authors have published

  • Appropriate use of statistics and treatment of uncertainties: In order to improve the inversion speed, the authors adopt the 2D Finite Differences method for forward modeling and take advantage of MPI parallel design strategy to realize parallel artificial fish swarm constrained inversion of 2D MagnetotelluricT data.

Conclusions: The clear objective of the article is to propose a new algorithm based on paralleling forward to effectively improve the inversion speed of 2D data. They show that the proposed investment method has good practicability.

Author Response

Thanks!

Reviewer 2 Report

1. The English writing is good, mostly, but really needs a careful review all the way as there are some long, unusual and confused sentences. For example, the sentence of L51-55 should be changed into past tenses.

2. In the Introduction, please add the cites of Liu et al., 2020 (DOI: 10.1002/gj.3627) and Sun et al., 2022 (Constrained Inversion of Audio Magnetotelluric for Identifying StrataA Case Study in Hami Basin. Earth Science, DOI: 10.3799/dqkx.2022.207), which are also recently constrained inversion cases of 2D MT.

3. Please move “Resistivity (Ohm.m)” to the color label in Fig.1 and add the physical quantity “Apparent resistivity” into the color label of Fig. 2. For all figures, it is better to add the physical quantity into the color label.

4. L156-157, “the parallel parallel scaling is 12.8 and 9.98, respectively”, remove a “parallel”?

5. As the method of artificial fish swarm algorithm has been well developed, I suggest the authors should simplify the section 3 and focus on 2D artificial fish swarm algorithm.

6. For the illustration of Fig. 10, it is better to change into “Apparent resistivity sections of the true model (a) and the model (b) from 2D inversion.”

7. What is the meaning of introducing Line 2 in Fig. 12, as there is no cite in the main text?

Author Response

  1. The English writing is good, mostly, but really needs a careful review all the way as there are some long, unusual and confused sentences. For example, the sentence of L51-55 should be changed into past tenses.

Reply: We have modified the sentence of L51-55.

  1. In the Introduction, please add the cites of Liu et al., 2020 (DOI: 10.1002/gj.3627) and Sun et al., 2022 (Constrained Inversion of Audio Magnetotelluric for Identifying Strata:A Case Study in Hami Basin. Earth Science, DOI: 10.3799/dqkx.2022.207), which are also recently constrained inversion cases of 2D MT.

Reply: We have added there references and modified all index of the reference.

  1. Please move “Resistivity (Ohm.m)” to the color label in Fig.1 and add the physical quantity “Apparent resistivity” into the color label of Fig. 2. For all figures, it is better to add the physical quantity into the color label.

Reply: We have modified figure 1-2, figure 6, figure 9-10, figure 12-14, added the physical quantity into the color label.

  1. L156-157, “the parallel parallel scaling is 12.8 and 9.98, respectively”, remove a “parallel”?

Reply: We have modified.

  1. As the method of artificial fish swarm algorithm has been well developed, I suggest the authors should simplify the section 3 and focus on 2D artificial fish swarm algorithm.

Reply: Thanks for the reviewers’ suggestion. We have modified.

  1. For the illustration of Fig. 10, it is better to change into “Apparent resistivity sections of the true model (a) and the model (b) from 2D inversion.”

Reply: We have modified.

  1. What is the meaning of introducing Line 2 in Fig. 12, as there is no cite in the main text?

Reply: We have deleted Line 2 in the figure.

Reviewer 3 Report

I think the manuscript contains not enough novel and original content as an independent research article since 2D magnetotelluric (MT) parallel-constrained inversion based on artificial fish swarm algorithms and 2D MT forward modeling by finite difference method with paralleling to frequency are not new.

Author Response

Reviewer 3: I think the manuscript contains not enough novel and original content as an independent research since 2D magnetotelluric (MT) parallel-constrained inversion based on artificial fish swarm algorithms and 2D MT forward modeling by finite difference method with paralleling to frequency are not new.

Reply: Thanks for the revierwer’s comments. The finite difference method with paralleling to frequency is a classical 2D MT forward method. The initial artificial fish swarm algorithm was proposed in 2003, which is not new. In 2015, we first applied the artificial fish swarm algorithm to 1D MT inversion. Now, we have developed the artificial fish swarm algorithm to 2D parallel-constrained MT inversion, which is an innovation compared with previous work. In addition, we have applied this method to field data processing and obtained good results. It shows that this method has a good application prospect.

Round 2

Reviewer 2 Report

The authors have revised the manuscript completely according to the reviewer’s comments. Thus, I suggest that this revised manuscript can be accepted. However, there are three points should revised.

1.      In the Eq. (2), the physical meanings of “ω” and “μ0” should be provided.

2.      In the Fig. 1, Fig. 6 and Fig. 9 as well as Fig. 13, the expression “Depth” is suggested to be revised as “Elevation” or the negative sign is suggested to be removed.

3.      In the main text such as the first sentence of section 4.1, the expression “Ω·m” should be changed as “Ω·m”, that is, the “m” should not be expressed by the italic.

Author Response

  1. In the Eq. (2), the physical meanings of “ω” and “μ0” should be provided.

Reply: We have added the physical meanings of “ω” and “μ0”.

  1. In the Fig. 1, Fig. 6 and Fig. 9 as well as Fig. 13, the expression “Depth” is suggested to be revised as “Elevation” or the negative sign is suggested to be removed.

Reply: We have modified Fig. 1, Fig. 6 and Fig. 9 as well as Fig. 13.

  1. In the main text such as the first sentence of section 4.1, the expression “Ω·m” should be changed as “Ω·m”, that is, the “m” should not be expressed by the italic.

Reply: We have modified “Ω·m” to “Ω·m”.
